# Bone Marrow Immune Microenvironment in Myelodysplastic Syndromes

**DOI:** 10.3390/cancers14225656

**Published:** 2022-11-17

**Authors:** Olga Kouroukli, Argiris Symeonidis, Periklis Foukas, Myrto-Kalliopi Maragkou, Eleni P. Kourea

**Affiliations:** 1Department of Pathology, University Hospital of Patras, 26504 Patras, Greece; 2Hematology Division, Department of Internal Medicine, School of Medicine, University of Patras, 26332 Patras, Greece; 32nd Department of Pathology, Attikon University Hospital, Medical School, National and Kapodistrian University of Athens, 12462 Athens, Greece; 4Department of Nutritional Sciences and Dietetics, School of Health Sciences, International Hellenic University, 54124 Thessaloniki, Greece; 5Department of Pathology, School of Medicine, University of Patras, 26504 Patras, Greece

**Keywords:** MDS, BM, microenvironment, HSC niche, pathogenesis, inflamm-aging, immune dysregulation, CHIP, immunosuppression, MDSC

## Abstract

**Simple Summary:**

The bone marrow (BM) microenvironment regulates normal hematopoiesis and exerts variable activity in various inflammatory, toxic, autoimmune, or neoplastic diseases and conditions. Most importantly, it has a major role in the pathogenesis of BM failure syndromes and particularly of myelodysplastic syndromes (MDS), in which it is dynamically implicated in the early phase of their pathogenesis, as well as in their course and evolution. This review presents a concise overview of the literature, highlighting the impact of the immune component of the BM microenvironment during normal hematopoiesis, in chronic inflammatory states, and in low- and high-risk MDS, laying the ground for further research on the cellular and biochemical immune parameters that participate in early stages of MDS pathogenesis and in disease evolution.

**Abstract:**

The BM, the major hematopoietic organ in humans, consists of a pleiomorphic environment of cellular, extracellular, and bioactive compounds with continuous and complex interactions between them, leading to the formation of mature blood cells found in the peripheral circulation. Systemic and local inflammation in the BM elicit stress hematopoiesis and drive hematopoietic stem cells (HSCs) out of their quiescent state, as part of a protective pathophysiologic process. However, sustained chronic inflammation impairs HSC function, favors mutagenesis, and predisposes the development of hematologic malignancies, such as myelodysplastic syndromes (MDS). Apart from intrinsic cellular mechanisms, various extrinsic factors of the BM immune microenvironment (IME) emerge as potential determinants of disease initiation and evolution. In MDS, the IME is reprogrammed, initially to prevent the development, but ultimately to support and provide a survival advantage to the dysplastic clone. Specific cellular elements, such as myeloid-derived suppressor cells (MDSCs) are recruited to support and enhance clonal expansion. The immune-mediated inhibition of normal hematopoiesis contributes to peripheral cytopenias of MDS patients, while immunosuppression in late-stage MDS enables immune evasion and disease progression towards acute myeloid leukemia (AML). In this review, we aim to elucidate the role of the mediators of immune response in the initial pathogenesis of MDS and the evolution of the disease.

## 1. Introduction

Normal hematopoiesis depends upon a tightly organized hierarchical system within the bone marrow. Pluripotent HSCs reside on the top of the BM hierarchy and have a self-renewal capacity that is exerted by their symmetrical division, while the asymmetric division of HSCs produces progenitor cells primed for differentiation into mature cells [1,2]. This system, besides ensuring the rapid production of hematopoietic cells, safeguards hematopoiesis from the introduction of mutations. Despite this structure, during HSC division, somatic mutations that provide a relative fitness advantage may occur. These mutations may dominate and give rise to clonal hematopoiesis with the potential of evolution into frank hematological malignancies [3]. MDS derive from such clonal, mutational events, which become more common with advancing age. However, extracellular events occurring also in the BM microenvironment, namely, persistent inflammatory stimulation, immune dysregulation, and increased oxidative stress, have also been associated with MDS development. The inability of MDS cells to engraft in murine models (that would propagate human MDS in experimental animals) has initiated the discussion about the potential pathogenetic or facilitating role of the BM microenvironment, upon which the survival and expansion of MDS stem cells are dependent [4]. In addition, alterations in cellular elements of the BM microenvironment appear involved in MDS pathogenesis, such as Dicer1 gene deletion in BM osteoprogenitor cells, which suffices to induce a myelodysplasia-like phenotype and strongly suggests the carcinogenetic effect of the stromal component of the HSC niche [5]. Herein, we review the involvement of the BM IME in physiologic conditions, in aging, in the generation of clonal hematopoiesis, and in the evolution of established MDS, to unravel its role in disease initiation and progression.

## 2. BM IME in Physiologic Conditions

### 2.1. Hematopoietic Cells in the HSC Niche during Homeostasis

The HSC niche constitutes an extrinsic regulator of self-renewal (S), multilineage differentiation (M), apoptosis (A), rest (R), and trafficking (T) of the HSCs, together known as the “SMART” properties of HSCs and consists of the extracellular matrix and various cellular components of the BM microenvironment [6]. Apart from non-hematopoietic cells, namely of osteoid lineage, perivascular, endothelial, fibroblastic, mesenchymal stromal, adipocytic, and neural cells [7], mature hematopoietic cells that normally reside in the bone marrow have a role in regulating hematopoiesis and therefore, they could be considered as constituents of the BM microenvironment [8].

Megakaryocytes (MKs) are localized adjacent to HSCs and restrain their proliferation by producing C-X-C motif ligand 4 (CXCL4), Transforming Growth Factor beta (TGF-β), and thrombopoietin (TPO) [9,10,11], [Figure 1]. Pinho et al. suggested that MKs selectively influence the proliferative capacity and the fate of platelet and myeloid-biased HSCs, characterized by the surface expression of the von Willebrand factor (vWF) [12].

CD150 expressing T regulatory cells (Tregs) reside in the HSC niche and maintain HSCs’ quiescence via the production of adenosine [13], [Figure 1]. In addition, Tregs generate an immune-privileged or immune-tolerant environment in the HSC niche, as becomes evident by their role in preventing the rejection of transplanted allogeneic hematopoietic stem/progenitor cells (HSPCs) [14].

Niche macrophages maintain long-term HSC (LT-HSC) quiescence, as they express Duffy antigen/receptor for chemokines (DARC) which binds to CD82 on LT-HSCs and leads to cell-cycle arrest [15], [Figure 1]. Consistently with the well-described erythroblastic islands with central “nutritional” macrophages, CD139+ macrophages promote normal hematopoiesis [16]. Additionally, they indirectly mediate HSC retention in the BM, by inducing C-X-C motif chemokine ligand 12 (CXCL12) secretion by stromal niche cells [17,18], [Figure 1]. In fact, depletion of the niche macrophages results in HSC mobilization in the periphery [19]. Alpha-smooth muscle actin (α-SMA)+ macrophages safeguard HSCs from exhaustion, by producing prostaglandin E2 (PGE2) and preserving low reactive oxygen species (ROS) levels in HSCs [20]. PGE2 additionally increases CXCL12 expression on the surface of stromal cells, thus preventing HSC to egress through marrow sinusoids into the circulation [21]. Dendritic cells (DCs) reside in the perivascular HSC niche and their ablation results in HSC mobilization, in part due to increased vascular permeability and CXC chemokine receptor 2 (CXCR2) signaling in the endothelium [22], [Figure 1].

### 2.2. Inflammatory Response of HSCs during Emergency Granulopoiesis

Emergency granulopoiesis is a compensatory physiological process in cases of systemic infection or inflammation that involves the increased proliferation of myeloid progenitor cells in the bone marrow, to replenish the rapidly consumed granulocytes in the periphery. Pathogen sensing, occurring either in the peripheral tissues or within the bone marrow, is the initiating event, which is mediated either by hematopoietic or by non-hematopoietic cells that express pattern recognition receptors (PRRs) [23]. HSPCs express Toll-like receptors (TLRs), a family of PRRs, which upon activation result in increased survival, proliferation, and myeloid differentiation of HSPCs [24]. TLR agonist stimulation of HSPCs leads to the secretion of hematopoietic growth factors such as Interleukin 6 (IL-6) [25].

Following inflammatory stimulation, HSC proliferative activity is only transiently increased [26] and their quiescence is sustained through the activation of purine-rich box1 (PU.1) transcription factor, that represses genes related to cell-cycle progression and protein synthesis [27].

The maintenance of HSCs’ quiescence is crucial for their self-renewal capacity [28]. Under normal conditions, HSCs only infrequently proliferate, and they would rather remain in a quiescent state. HSCs display kinetic heterogeneity with a “dormant” but potent population and a relatively more frequently dividing population, which gradually loses its self-renewal potential and becomes susceptible to DNA damage [29].

## 3. BM IME in Aging

Impairment of the immune and hematopoietic system in aged individuals has been well-described and can be manifested, for example, by a weak immune response following vaccination, a higher incidence of autoimmunity, the generation of anemia and of non-specific hematopoietic failure and by an increased risk of hematologic malignancies. The impairment is at least partly explained by cell-intrinsic mechanisms, including elevated intracellular oxidative stress and DNA damage, telomere shortening, and various epigenetic changes [30]. However, extrinsic signals of the BM microenvironment, in particular inflammatory signals, may be the initiators of such intrinsic HSC defects [31].

### 3.1. Myeloid Skewing

Contrary to the former belief that immune memory is an exclusive feature of adaptive immunity, it is now recognized that epigenetic changes in macrophages, monocytes and NK cells are responsible for the “trained immunity” of the innate immune system [32]. Long-lived cells and even HSCs are also susceptible to epigenetic immune memory [33]. HSC exposure to lipopolysaccharide (LPS), besides an acute response, establishes persistent open chromatin regions (OCRs), corresponding to genes of myeloid lineage commitment and primes HSCs for an accelerated myelopoiesis upon secondary exposure [34].

In mice, aging is accompanied by increased platelet and myeloid cell counts and reduced numbers of red blood cells and lymphoid cells, which reflects the myeloid-skewing of HSCs in the BM [35,36,37,38]. In fact, while erythropoiesis and lymphopoiesis are compromised in aged individuals, myelopoiesis is maintained [39]. The pattern of the expression of specific markers, namely the overexpression of CD150 and the downregulation of CD68, that characterize the myeloid-biased phenotype of HSCs is noted particularly among aged HSCs [40,41]. The composition of multipotent LT-HSCs shifts with age and myeloid-biased LT-HSCs (mLT-HSCs) prevail and preferentially expand during an inflammatory challenge. The continuous and cumulative exposure to various inflammatory challenges during a lifetime may account for the myeloid bias of aging. CD61 has been identified as a marker of mLT-HSCs and characterizes a population of LT-HSCs primed to respond to inflammatory stimuli. Specific transcription factors, namely Krupper-like factor 5 (Klf5), Ikaros family zinc finger protein 1 (Ikzf1), and Signal transducer and activator of transcription 3 (Stat3), have been identified as regulators of the inflammatory myeloid bias in aged LT-HSCs [42]. Quantitative analysis of committed hematopoietic progenitors in the human BM reveals a decrease in common lymphoid progenitor (CLP) cells in the elderly, while common myeloid progenitor (CMP) cells are maintained [43].

### 3.2. Functional Decline of Aged and Overstimulated HSCs

Aged HSCs divide symmetrically, as opposed to asymmetric divisions of young HSCs [44]. Following repeated stimulation by inflammatory challenges, HSCs exhibit a functional decline and inability to regenerate. Transient exposure to polyinosinic:polycytidylic acid (pI:pC) induces Interferon alpha (IFN-α) response and proliferation of HSCs. Conversely, the repeated challenge of pI:pC injection resulted in a compromise of the proliferative potential of LT-HSCs, while no substantial replenishment of functional HSCs was observed after a recovery period [45,46]. Sustained LPS exposure and TLR4 stimulation impinge on HSC competitive repopulation capacity [47].

The HSC niche may contribute to the functional decline of HSCs, as it is implied by transplantation studies, according to which, the myeloid skewing of aged HSCs, transplanted into young recipients was reversed [48]. The stromal compartment in aged BM acquires inflammatory transcriptional profiles. Interleukin 1 beta (Il1β) gene is strongly expressed and potentially relevant to age-related HSC defects, given the effect of IL-1β in driving myeloid skewing and suppressing the self-renewal capacity of HSCs [49]. The complement system, involved in the mobilization of HSPCs in the periphery during stress hematopoiesis, is also activated [50].

### 3.3. Inflamm-Aging

In the elderly, a chronic, mild inflammatory activity, termed “inflamm-aging” is associated with age-related diseases [51,52]. Increased number of pro-inflammatory cytokines, including Regulated upon Activation, Normal T Cell Expressed and Presumably Secreted (Rantes) [48], IL-1β, Interleukin 6 (IL-6) [36], TGF-β [37], and Tumor Necrosis Factor-alpha (TNF-α) [53], are present in the bone marrow of aged individuals. The pro-inflammatory cytokines in the senescent BM mediate myeloid skewing and HSC impairment [Figure 2]. Even in non-hematologic diseases, such as rheumatoid arthritis, the observed myeloid-biased hematopoiesis is attributed to the elevated numbers of pro-inflammatory cytokines in this disease [54].

Interleukin 1 (IL-1) promotes myeloid differentiation via Nuclear Factor kappa-light-chain enhancer of activated B cells (NF-κB) signaling and downstream PU.1-mediated gene expression but chronic IL-1 stimulation impairs the self-renewal capacity of HSCs [49]. Transient exposure to TNF delivers pro-survival messages to HSCs; however, persistent exposure results in NF-κB inactivation and HSC necroptosis [55]. Aging-related TNF-α overproduction upregulates Interleukin-27 receptor alpha (IL-27Rα), a sensor of inflammation on HSCs, and is associated with their functional decline. IL-27Rα+ HSCs exhibit reduced reconstitution capacity, myeloid bias, and overexpression of inflammatory genes [53]. IFN-α favors HSC exit from quiescence and inflicts proliferation-induced genotoxic stress [46].

While NF-κB baseline activity is indispensable for the preservation of hematopoietic homeostasis [41], increased NF-κB signaling during aging predisposes to HSC differentiation, limiting their self-renewal capacity. Chronic inflammatory stimulation of young mice via mechanisms that involve NF-κB activation also induces myeloid differentiation and recapitulates the effects of aging [56].

### 3.4. Molecular Association of Aging and Inflammation

Aging and inflammation are variously intertwined on the molecular level [57], [Figure 2]. Cytosolic DNA fragments from incomplete DNA repair activate the cyclic guanosine monophosphate-adenosine monophosphate synthase (cGAS)-stimulating interferon gene (STING) aging pathway, that leads to Interferon beta (IFN-β) production [58]. Intracellular double-stranded DNA (dsDNA) from double DNA breaks leads to absent in melanoma 2 (AIM2) inflammasome assembly, which enables the proteolytic activation by Caspase-1 and secretion of IL-1β and Interleukin 18 (IL-18) [59]. The epigenetic dysregulation of aging results in the upregulation of inflammation-associated genes in HSCs [60]. Moreover, the compromised autophagy of aging and mitochondrial stress activates the NOD-, LRR-, and pyrin domain-containing protein 3 (NLRP3) inflammasome-Caspase-1 cascade, exacerbating the inflammation via cytokine production and induce pyroptosis of HSCs [61,62]. Pyroptosis and necroptosis are two well-established inflammatory forms of cell death that are activated during aging in addition to traditional apoptotic pathways. Necroptosis is programmed cell death, following activation of tumor necrosis factor receptors (TNFRs), TLRs and interferon receptors (IFNRs), and downstream mediators, such as receptor-interacting protein kinase 1 (RIPK1), receptor-interacting serine/threonine kinase 3 (RIPK3), and mixed lineage kinase domain-like (MLKL) that leads to the release of damage-associated molecular patterns (DAMPs) [63]. Similarly, pyroptosis is activated by inflammatory signals and involves inflammasome assembly and caspase-1 activation, which result in IL-1β and IL-18 secretion [64].

### 3.5. Alterations in Immune Cells

The myeloid skewing of advanced age is accompanied by alterations of immune cells in the BM [Figure 2]. Increased natural killer (NK) cells in the senescent BM microenvironment restrain effective B lymphopoiesis by downregulating the transcription factor E47 in B cell precursors in a TNF-α dependent way [65]. Plasma cells accumulate in the bone marrow of aged mice, contribute to myelopoiesis, and acquire a TLR-responsive gene signature [8]. Age-associated B cells (ABCs) are mature B cells that contribute to the inhibition of B lymphopoiesis via TNF-α secretion in the pro-inflammatory BM microenvironment of aged mice [66]. Aged macrophages in the BM induce platelet bias in HSCs via IL-1β [38].

## 4. BM IME in Clonal Hematopoiesis

### 4.1. Clonal Hematopoiesis (CH) in Advanced Age

During chronic immune stimulation, the increased proliferation rate renders HSC statistically more prone to genetic errors [67]. Different studies of DNA-sequencing analysis on many thousands of people, unselected for hematologic malignancies, identified recurrent mutations in leukemia-driver genes, such as DNA-methyltransferase 3A (*DNMT3A*), Tet methyl cytosine dioxygenase 2 (*TET2*), and additional sex combs like 1 (*ASXL1*), whose prevalence increased with age and was associated with an increased risk of hematologic neoplasia and overall mortality [68,69,70]. Clonal Hematopoiesis of Indeterminate Potential (CHIP) was subsequently defined as the clonal expansion of hematopoietic cells upon acquisition of a somatic mutation in leukemia-associated driver genes at a variant allele fraction of at least 2% and in the absence of diagnostic criteria for another hematologic condition [71]. Intrinsic defects of HSCs and selective pressure mechanisms in aging may explain the generation of clonal hematopoiesis [3]. Replication-associated DNA damage in aged HSCs triggers apoptosis or differentiation and possibly predisposes to further mutagenesis in a fitness landscape where HSCs that retain self-renewal capacity are the ones selected to expand [72]. Sensibly, mutations in epigenetic modifiers *DNMT3A*, *TET2*, and *ASXL1*, that may enhance self-renewal or introduce a differentiation blockage are often present in CH [3]. Recent data, however, suggest that mutations linked to clonal hematological disorders are not newly acquired events at the time of diagnosis, but instead precede the clinical manifestation by many years or even decades. Interestingly, *DNMT3A* mutation has been found to occur already in utero during embryogenesis or during childhood and to slowly grow during a lifetime [73]. Therefore, it is not just the acquisition of mutations but the fitness advantage of the mutated clone in the inflammatory context of these diseases that need to be investigated. To this effect, there is in vitro evidence that *TET2* mutations facilitate clonal expansion in aging, by providing TNF-a resistance [74].

### 4.2. Dysregulation of Inflammatory Pathways and Competitive Advantage of Mutant HSC

In clonal hematopoiesis and MDS, various dysregulated molecules in mutant HSPCs affect supramolecular organizing centers (SMOCs); thus, major signaling pathways that maintain HSC survival, function and inflammatory response are disrupted. SMOCs are the protein complexes of receptors, adaptors, and effector molecules, following PRR activation and include myddosome and trifosome complex, inflammasome, and necroptosome [75]. TET2 mutation is linked to the overexpression of myddosome’s molecule TRAF6 [76], as well as the activation of NLPR3 inflammasome [77]. According to the suggested model of MDS initiation, a clonal event, for example, the acquisition of DNT3A, TET2 or ASXL1 mutation and the subsequent SMOC dysregulation, provide a competitive advantage to HSCs in the context of chronic inflammation and increased exposure to cytokines and alarmins in the bone marrow, derived from aging, chronic infection, or immune dysregulation. The selective advantage of mutant HSCs stems from their altered response to inflammation. MDS HSCs preferentially exhibit noncanonical activation of NF-κB, over the canonical NF-κB activation of normal HSCs and inhibition of noncanonical NF-κB signaling abrogates their competitive advantage [75,76].

### 4.3. Evidence of Inflammatory Overstimulation in CHIP

The association between CHIP and cardiovascular disease (CVD) is well-established [70,78,79] and inflammation constitutes a plausible link. Inflammation contributes to CVD [80] while CHIP emerges under inflammatory stress and augments inflammation by increasing the release of pro-inflammatory cytokines which, through a feedback loop, aid clonal expansion [81,82]. *TET2* loss of function promotes atherosclerosis due to NLRP3-mediated IL-1β secretion by macrophages [83]. Patients with *DNT3A*-CHIP-driver mutation exhibit a pro-inflammatory T-cell distribution as evidenced by a significant increase in the T helper 17 (Th17) cells/Tregs ratio [84].

## 5. BM IME in MDS

### 5.1. 5q− Syndrome as a Representative Example of Immune Dysregulation in MDS Initiation

5q− syndrome, a common MDS subtype, provides evidence for the overstimulation of the innate immune system in the development of myelodysplasia. Deletion of miR-145 and miR-146, located on chromosome 5q, phenocopies 5q− syndrome, and activates NF-κB signaling, since miR-145 and miR-146 target NF-κB pathway mediators, Toll-interleukin-1 receptor domain-containing adaptor protein (TIRAP), and Tumor necrosis factor receptor-associated factor 6 (TRAF6), respectively. The dysplastic phenotype in TRAF6-transduced mice is linked to IL-6 overexpression and paracrine action, while clonal expansion and acute myeloid leukemia (AML) development rely upon TRAF6 activation in a cell-autonomous fashion [85]. Interestingly, lenalinomide, the treatment of choice for 5q− syndrome, has been found to suppress IL-6 expression [86]. Aged mice show decreased miR-146 expression and elimination of miR-146 results in premature HSC aging and development of myeloid malignancy, dependent upon IL-6 and TNF activation. When IL-6 and TNF were targeted, the function of HSCs in miR-146-deficient mice was restored [87].

### 5.2. Overactivation of Inflammatory Pathways in MDS HSCs

In MDS, emerging data support cell-intrinsic mechanisms in the dysregulated innate immune and inflammatory signaling, involved in disease pathophysiology [75]. MDS HSCs exhibit increased expression of TLR receptors [88]. TLR-4 upregulation correlates with increased apoptosis in MDS [89]. TLR-2 overexpression is linked to low-risk disease while TLR-6 overexpression characterizes high-risk MDS. Chronic stimulation of the TLR-2/TLR-6 complex accelerates progression to AML [90]. The overstimulation of the NLRP3 inflammasome by DAMPs, most notably S100 calcium-binding protein A8 (S100A8) and S100 calcium-binding protein A9 (S100A9), and caspase-1 activation is a central event in MDS, irrespective of the underlying mutational profile [91,92]. In addition, overactivation of the TGF-β signaling, via the upregulation of Mothers against decapentaplegic homolog/SMAD family member 2 (SMAD2) or the inhibition of the negative regulator Mothers against decapentaplegic homolog/SMAD family member 7 (SMAD7), correlates with myelosuppression in MDS [93,94]. Certain driver mutations in MDS, especially in splicing factors, contribute to the dysregulation of immune-related molecules. Mutated U2 small nuclear RNA auxiliary factor 1 (U2AF1) produces a long isoform of Interleukin-1 receptor-associated kinase 4 (IRAK4) (IRAK4-L), that overactivated NF-κB signaling via myddosome assembly [95]. Splicing factor 3B subunit 1 (*SF3B1*) mutations, one of the most encountered mutations in MDS, also result in IRAK4-L overexpression, but additionally hinder Mitogen-activated protein kinase kinase kinase 7 (MAP3K7) expression, which indirectly restricts NF-κB signaling, leading to hyperactivated NF-κB signaling. Serine and arginine rich splicing factor 2 (SRSF2) mutations lead to aberrant splicing of caspase 8, whose inhibition is necessary for the induction of necroptosis instead of apoptosis [96].

### 5.3. The Immunomodulatory Role of High Mobility Group Box-1 (HMGB1) in MDS BM

HMGB1 is a nuclear, non-histone DNA-binding protein that regulates chromatin remodeling and transcriptional states. HMGB1 belongs to endogenous DAMPs, known as alarmins, and perpetuates inflammation following its passive release from necrotic cells or active secretion by innate immune cells or “stressed” cells. NF-kB activation is involved in HMGB1 signaling, subsequently to HMGB1 ligation to both advanced glycation end products receptor (RAGE) and TLRs [97]. In the BM, HMGB1 promotes HSC self-renewal and differentiation capacity but is also involved in hematologic malignancies [98]. In MDS, HMGB1 emerges as an important and potentially targetable contributor to the inflammatory BM milieu. Serum HMGB1 levels are significantly increased in MDS patients when compared to healthy controls [99,100] as well as compared to patients with other BM failure syndromes, highlighting its specific role in MDS pathophysiology [100]. Circulating HMGB1 levels are higher in low-risk MDS than in high-risk MDS patients, in consistence with the greater BM inflammatory load of the former group [100]. The defective apoptotic cell clearance by macrophages in the BM of MDS patients may secondarily initiate cell necrosis and release of HMGB1, which then leads to TLR4 activation and cytokine secretion [99]. Inflammasome activation also results in HMGB1 release [101], which may constitute an additional mechanism of HMGB1 overexpression in MDS. Kam et al. reported a two-to-three-fold higher expression of HMGB1 in MDS CD34+ cells compared to normal hematopoietic cells. Most importantly, they found that HMGB1 inhibition diminishes MDS cell expansion and has an additive effect on azacytidine or decitabine treatment while sparing normal hematopoietic cells. HMGB1 inhibition also correlates with downregulation of TLR receptors and NF-kB signaling, which suggests the immunodulatory role of HMGB1 in the BM [102].

### 5.4. Alterations of Cytokines and Immune Cells in the BM Microenvironment

#### 5.4.1. Cytokines

Increased levels of serum TNF-α, TGF-β, IL-6, and Interleukin 8 (IL-8) have repeatedly been reported in MDS [103,104,105]. Cytokine levels reflect the distinct immune dysregulation mechanisms that are involved in different disease stages. Low-risk MDS is characterized by an increased apoptotic rate and a pro-inflammatory microenvironment, where type I cytokines (e.g., IL-1β, Interleukin 7 (IL-7), IL-8, Interleukin 12 (IL-12)) dominate [106], [Figure 3]. On the contrary, high-risk MDS patients exhibit comparatively decreased apoptosis and clonal expansion of malignant cells due to immune evasion that is mediated by inhibitory factors, namely Interleukin 10 (IL-10) and soluble Interleukin-2 Receptor (sIL-2R) [107,108], [Figure 4]. The number of CD8+ T cells and IL-10 levels are inversely related [109].

Apoptosis is one of the most notably activated pathways in CD34+ cells from refractory anemia (RA) patients, when compared to healthy controls or refractory anemia with excess blast (RAEB) patients [110,111], [Figure 3]. The extrinsic apoptotic pathway involves the activation of the Fas death receptor (Fas) on the cell surface by Fas ligand, which is overexpressed in MDS patients [112]. Whereas normal CD34+ progenitor cells are devoid of Fas, its expression is induced by cytokines such as TNF-α and Interferon gamma (IFN-γ), which are elevated in low-risk MDS patients, when compared to high-risk MDS patients [113,114]. TNF-α selectively stimulates TNF receptor 1 (TNFR1), leading to the activation of proinflammatory signaling pathways and apoptosis, as opposed to TNF receptor 2 (TNFR2), which has an anti-apoptotic function [115]. In line with the pro-apoptotic microenvironment of low-risk MDS and the anti-apoptotic features of high-risk MDS, BM cells in RA patients overexpress TNFR1 [Figure 3], while TNFR2 expression is notable in RAEB patients [Figure 4], [116]. High serum levels of TNF-α in MDS patients are correlated with poorer performance status, higher leukocyte counts, β2-microglobulin and creatinine levels and on the contrary, lower TNF-α levels have been associated with favorable prognosis [117]. In advanced MDS, high serum levels of IL-6 and Granulocyte-macrophage colony-stimulating factor (GM-CSF) are observed. IL-6, IL-7, and C-X-C motif chemokine ligand 10 (CXCL10) may serve as independent prognostic factors of survival [118].

#### 5.4.2. Immune Cells

##### Lymphocytes

A subset of MDS patients exhibit the clonal expansion of T cells, characterized by T cell receptor (TCR) Vβ-repertoir skewing, probably reflecting an antigen-driven process [119]. Immunosuppressive therapies reverse the TCR-Vβ phenotype of T cells in hypocellular MDS [120]. A CD8+/CD57+/CD28− T cell phenotype is overrepresented in MDS patients compared to age-matched controls [121,122,123] and characterizes a population of terminally differentiated, memory/effector T cells that have undergone repeated cell division cycles [124]. Natural killer group 2D (NKG2D) and CD244, which are normally NK receptors, are expressed on MDS T cells and render effector capacity while HSCs express the CD244 ligand, CD48 [121], [Figure 3]. Diminished expression of the lymph node homing receptors C-C chemokine receptor 7 (CCR7) and L-Selectin (CD62L), also found on clonal MDS T cells, coincides with lymphocyte expansion in the bone marrow [121]. CD48 expression on MDS HSCs is further augmented following lenalinomide treatment, which may represent a mechanism of induction of apoptosis by this agent [125]. Deprivation of CD8+CD57+ T cells in vitro triggers colony formation of BM mononuclear cells in a subset of MDS cases. In the cases with abnormal karyotypes (+8, 20q−, 5q−) the proportion of clonal cells increases significantly [126]. These findings support the inhibitory effect of CD8+ cells on the neoplastic cells of MDS.

Potential epitopes on MDS stem cells that activate anti-tumoral CD8+ T cells include Wilms tumor 1 protein (WT1), cancer-testis antigens (CTAs), proteinase 3, and Major Histocompatibility Complex class I (MHC I) [127,128,129]. Alongside malignant stem cells, CD8+ T cells target non-malignant hematopoietic cells and inhibit hematopoiesis in MDS. CD34+ cells and mononuclear cells of trisomy 8 MDS patients overexpress WT-1, which triggers the clonal expansion of WT1-specific CD8+ cells that contribute to myelosuppression. These findings are consistent with the increased efficacy of immunosuppressive therapy in trisomy 8 MDS patients [128]. Hypomethylating agents (HMAs) significantly upregulate the expression of CTAs on AML leukemic cells, which correlates with the enhanced recognition of tumor cells by T cells [130].

The absolute number of CD4+ T cells is decreased in MDS patients compared to healthy controls and results in a reduction in the CD4/CD8 ratio [131]. The abnormal CD4/CD8 ratio in young MDS patients further correlates with response to immunosuppressive therapy and probably reflects the loss of the regulatory T cell (Treg) compartment of CD4+ cells. A higher proliferative T cell index in these patients may reflect activation of “homeostatic proliferation”, mediated by Interleukin 2 receptor gamma (IL-2Rγ), leading to non-specific T cell expansion, including self-reactive T-cells [132]. This model serves as a potential alternative mechanism of CD8+ expansion, besides antigen-specific responses, and may explain the autoimmunity-associated features of MDS [132,133]. Moreover, in low-risk MDS patients Th17 cells are increased and Tregs are inversely decreased and functionally impaired [101]. In high-risk MDS patients, Tregs are expanded while CD8+ T cells and NK cells are decreased [134,135,136]. Symeonidis et al. have associated CD3+ and CD8+ cell lymphopenia with increased risk of infections, transformation to AML, as well as decreased overall survival. According to this theory, although in MDS recognition of autologous antigens is impaired favoring autoimmunity, the immune reaction against CD34+ cells is increased, as compared to normal subjects. Thus this in vitro finding, together with the increase in suppressive CD8+ cells, might be considered as a defensive mechanism of the host against the expanding MDS clonal cells, which ultimately is abrogated during disease progression to higher-risk disease or AML, generating immune tolerance against the expanding population of blast cells [137].

The conversion to an immune evasive microenvironment in advanced MDS is facilitated by the overexpression of immune checkpoint molecules, including the programmed cell death of protein 1 (PD-1), its ligand PD-L1, and cytotoxic T lymphocyte-associated protein 4 (CTLA-4), leading to T cell exhaustion [138], [Figure 4]. The interaction of PD-1 and PD-L1 suppresses TCR-mediated proliferation and cytokine expression [139]. High-risk MDS patients show increased PD-L1 expression on blast cells, compared to healthy controls [140]. Yang et al. observed at least a two-fold upregulation of PD-L1 on BM CD34+ cells in 36% of studied samples from MDS patients and an enhanced expression of PD-L1, PD-1 and CTLA-4 after HMA treatment [141]. Secreted factors in the microenvironment, such as IFN-γ and TNF-α, upregulate PD-L1 expression [142,143] and recently, the role of S100A9, a central molecule in the MDS pathophysiology, as an inducer of PD-1 and PD-L1, has been detected [144]. Contrary to former knowledge, recently, Ferrari et al. did not observe an inhibition of cytotoxicity after de novo PD-L1 expression on MDS cells [145]. Additional inhibitory signals originating from the innate immune system may interfere with anti-tumor immunity.

##### Macrophages

Macrophage-mediated apoptosis in low-risk MDS affects myeloid progenitor cells expressing the prophagocytic receptor calretinin (CRT) rather than HSCs, which lack CRT expression and are rescued from apoptosis [146], [Figure 3]. Myeloid progenitor cells from high-risk MDS patients additionally overexpress CD47, which is a surface anti-phagocytic molecule abnormally expressed on tumor cells that binds to its receptor Signal regulatory protein alpha (SIRPa) on macrophages and inhibits phagocytosis, counteracting CRT [146], [Figure 4]. Despite the increase in monocyte cells in the peripheral blood of MDS patients, decreased differentiation into macrophages and attenuated phagocytic capacity of macrophages are reported [147].

Aside from the impaired recognition and phagocytosis of MDS cells by macrophages [147], Zhang et al. found an abnormal polarization of macrophages with decreased M1:M2 ratio in high-risk MDS patients [148]. M1 macrophages participate in antitumor immune response via secretion of pro-inflammatory factors and recognition of tumor-associated antigens (TAAs) [149]. However, M1 macrophages derived from MDS patients show decreased expression of IL-1β and TNF-α [148]. Tumor-associated macrophages (TAMs) in late-stage cancer are characterized by an M2-like phenotype (low IL-12 and high IL-10 expression) and potentiate tumor progression through decreased tumoricidal activity, angiogenesis, and matrix remodeling [150]. The increase in the M2 macrophages in high-risk MDS patients may therefore contribute to impaired immunosurveillance and clonal expansion, in line with studies highlighting the role of TAMs in AML progression [151,152] and the promising results of macrophage repolarization [153].

##### NK Cells

The cytolytic function of NK cells is compromised in MDS patients [154], partly due to the reduced expression of the stimulating receptors DNAX accessory molecule 1 (DNAM-1) and NKG2D on the surface of NK cells, while their expression is inversely correlated to the BM blast counts [155]. Hejazi et al. observed a significant reduction in NK cell numbers preferentially in high-risk MDS patients and correlated their functional deficiency to low levels of granzyme B and perforin as well as the NK cells’ immaturity in MDS [156], [Figure 4]. The inability of NK cells to fully mature may stem from their ineffective support from MDS mesenchymal stromal cells (MSCs) [157], since stromal support is a prerequisite for the development of mature, Killer-cell immunoglobulin-like receptor (KIR)-expressing NK cells [158]. Partial clonal involvement of NK cells [159] and certain congenital mutations that link MDS predisposition and NK cell alterations [160,161,162] highlight the contribution of potential intrinsic defects to NK cell impairment.

##### Dendritic Cells

Lower numbers and functional deficiency of DCs in MDS may contribute to immunodeficiency and autoimmunity of MDS. FISH analysis revealed the origin of MDS DCs from the malignant clones [163]. CD34+ progenitor cells from MDS patients exhibit impaired in vitro generation of DCs [164]. Similarly, the differentiation of blood monocytes from MDS patients to DCs is impaired. Immature monocyte-derived DCs (MoDCs) show diminished expression of surface molecules CD80 and CD1a, reduced endocytic capacity and ineffective maturation following TNF-α stimulation [165]. The ineffective induction of T-cells [166] and altered cytokine secretion (less IL-12 and more IL-10 expression) [167] are further aspects of DC impairment in MDS. MDS DCs show downregulation of transcripts involved in pro-inflammatory pathways which may explain their reduced immune responsiveness [168].

##### MDSCs

MDSCs are immature immune cells increased in the context of immunosuppression, inflammation, and cancer [169] and contribute to cancer progression [170]. The two-stage model of MDSC involvement in cancer requires firstly the expansion of myeloid cells under chronic inflammatory stimulation and secondly their activation in the tumor microenvironment [171]. Rationally, MDS BM inflammatory microenvironment facilitates MDSC development [172]. Elevated numbers of MDSCs are found in the BM of MDS patients, mostly high risk, and correlate with Treg expansion and disease progression [173]. MDSCs serve as a source of inhibitory cytokines, such as IL-10, TGF-β, NO, and arginase [Figure 4] and contribute to the suppression of hematopoiesis in MDS and the induction of T cell tolerance [174]. MDSCs expand also in the bone marrow of aged healthy individuals, parallel to the rise of inflammatory cytokines in the serum, at least in part due to NF-κB activation [175,176]. However, MDS-derived MDSCs overexpress CD33, and their suppressive function depends upon the binding of pro-inflammatory molecules, such as S100A9, to the CD33 receptor on MDSCs [174,177], [Figure 4]. Forced MDSC expansion in S100A9 transgenic mice suffices to induce human MDS phenotype, while induced terminal differentiation eliminated MDSCs and restored hematopoiesis. A mechanism in which CD33 overexpression overrides maturation signals from Immunoreceptor tyrosine-based activated motifs (ITAM)-associated receptors and retains the expansion of immature MDSCs was proposed [174]. High levels of IL-6 and chemokine (C-C motif) ligand 2 (CCL2) in MDS patients may additionally induce MDCSs, as these molecules activate the STAΤ3 pathway, which is crucial for MDSC-mediated inhibition of CD8+ T cell function [178]. Tao et al. found that CD8+ T-cell exhaustion in MDS was partially dependent upon activation of the T-cell immunoglobulin and mucin-domain containing 3 (TIM3)/Galectin 9 (Gal 9) pathway [Figure 4] and suggested that Gal 9 upregulation by MDSCs is responsible for this activation [179]. TIM3/Carcinoembryonic antigen-related cell adhesion molecule 1 (CEACAM1) pathway activation may also be an alternative mechanism of MDSC involvement in T-cell exhaustion [180]. Irrespective of the specific mechanisms of MDSC-induced inhibition, MDSCs do not harbor the same mutations as the dysplastic clones [174], thus constituting an independent cellular component of the microenvironment that is actively involved in MDS pathophysiology.

### 5.5. Role of MSCs in the Inflammatory BM Milieu

MSCs are a stromal component of the MDS microenvironment that may be involved in the pathogenesis and evolution of MDS via inflammatory signaling [181]. MSCs functionally decline with age [182]. In MDS, MSCs acquire distinct genetic alterations from the malignant clone [183] and show activation of inflammatory pathways [184,185]. Importantly, MSCs secrete S100A8/9, which generates genotoxic stress in HSCs [186], [Figure 4]. S100A8/9 expression in MSCs, associated with activation of p53 and TLR pathways, is predictive of leukemic evolution in MDS [186]. MSCs have different immunomodulatory effects depending on the disease stage. They suppress DC maturation and function more potently in high-risk MDS compared to low-risk MDS, and this suppression is mediated through TGF-β1 secretion by MDS MSCs [187], [Figure 4]. Sarhan et al. attributed the immunosuppressive properties of MDS MSCs, specifically the inhibition of NK cell function and T-cell proliferation, on the induction of suppressive monocytes [188].

## 6. Therapeutic Implications

The emerging knowledge on MDS IME pathology identifies various potentially targetable molecules for novel therapeutic options that could complement traditional treatment or act as salvage therapy. Agents targeting several of the aforementioned components of the BM IME have been developed and are under clinical trials. Magrolimab, for example, is an anti-CD47 monoclonal antibody that counteracts the inhibition of phagocytosis in MDS [189] and potentially synergizes with azacytidine [190]. The efficacy of novel anti-TIM3 antibodies in combination with azacytidine is tested in phase Ib studies [191,192]. High-risk MDS patients may also benefit from combination therapy of anti-PD-1 inhibitor pembrolizumab with azacytidine [193]. The overexpression of TLRs in MDS provides the rationale for the development and interrogation in clinical trials of the anti-TLR2 antibody OPN-305 [194]. The above represents only a fraction of the factors, that immunotherapy could target to modulate the BM IME in MDS [195].

## 7. Conclusions

The overstimulation of inflammatory processes in the BM and inflamm-aging lead to functional impairment of HSCs and paves the way for clonal hematopoiesis and MDS initiation. Inflammatory signaling is rewired and confers a selective advantage to HSCs. The IME mediates BM failure and is dynamically involved in MDS pathophysiology, evolving from a pro-inflammatory and pro-apoptotic into a permissive milieu that enables immune evasion, clonal expansion, and disease progression. Our increasing understanding of the intricate mechanisms of the IME’s involvement serves to highlight potential targets for immunotherapy in MDS.

## Figures and Tables

**Figure 1 cancers-14-05656-f001:**
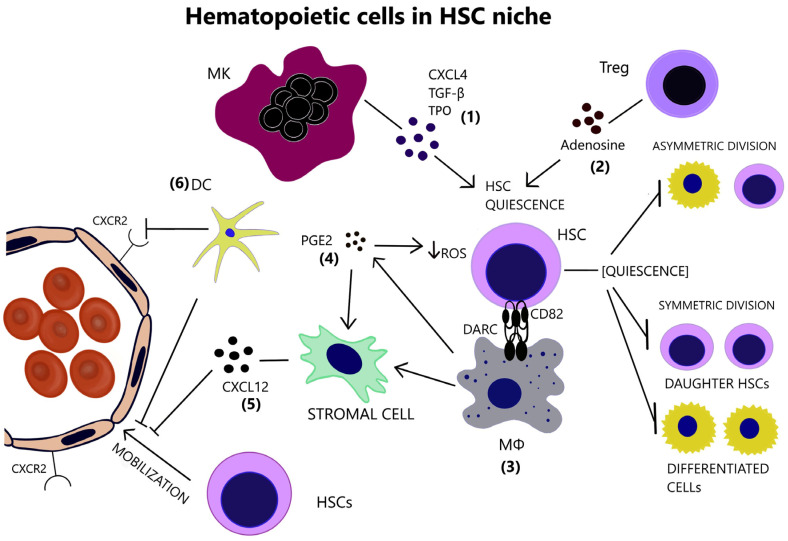
(**1**) Megakaryocyte support the preservation of HSC quiescence, through the secretion of CXCL4, TGF-β and TPO. (**2**) Tregs also contribute to HSC quiescence by generating adenosine, which protects HSCs from oxidative stress. (**3**) Macrophages express DARC, whose interaction with CD82 on the surface of HSCs restrains their proliferative activity. (**4**) a-SMA(+) macrophages are capable of COX-2-mediated PGE2 production that maintains low ROS levels in HSCs. (**5**) Macrophages additionally hinder HSC mobilization through the induction of CXCL12 by stromal cells. (**6**) Dendritic cells further aid HSC retention in the BM via regulation of vascular permeability and endothelial CXCR2 signaling. HSC inactive, quiescent state inhibits self-renewal and differentiation. (CXCL4 = C-X-C motif ligand 4, CXCL12 = C-X-C motif chemokine ligand 12, CXCR2 = CXC chemokine receptor 2, DARC = Duffy antigen/receptor for chemokines, DC = Dendritic cell, HSC = Hematopoietic stem cell, MK = Megakaryocyte, MΦ = Macrophage, PGE2 = Prostaglandin E2, ROS = Reactive oxygen species, TGF-β = Transforming growth factor beta, TPO = Thrombopoietin, Treg = T regulatory cell).

**Figure 2 cancers-14-05656-f002:**
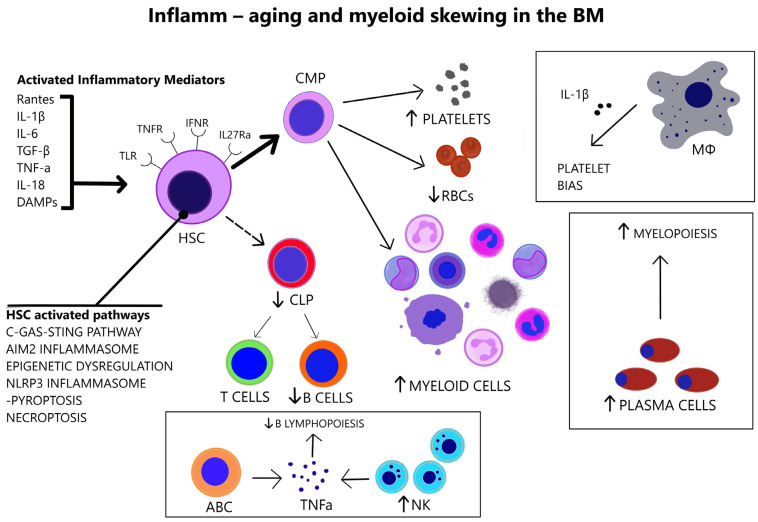
In the aged BM, there is increased number of pro-inflammatory cytokines that act on HSC receptors (TLRs, TNFRs, IFNRs, IL-27Rα). Various abnormally activated pathways in HSCs (listed above) link HSC senescence and BM inflammation. The overstimulated HSCs are myeloid-biased. Immune cell alterations in the senescent BM are consistent with myeloid skewing. Aged macrophages drive platelet bias via IL-1β. Plasma cells are increased and induce myelopoiesis. Increased NK cells and ABCs in the aged BM lead to TNFα-mediated impairment of B lymphopoiesis. (ABC = Age-associated B cell, AIM2 = Absent in melanoma 2, C-GAS-STING = Cyclic guanosine monophosphate-adenosine monophosphate synthase—Stimulating interferon gene, CLP = Common lymphoid progenitor, CMP = Common myeloid progenitor, DAMPs = Damage-associated molecular patterns, HSC = Hematopoietic stem cell, IFNR = Interferon receptor, IL-1β = Interleukin-1 beta, IL-6 = Interleukin-6, IL-18 = Interleukin-18, IL-27Rα = Interleukin-27 receptor alpha, MΦ = Macrophage, NLRP3 = NOD-, LRR- and pyrin domain-containing protein 3, NK = Natural killer, TGF-β = Transforming growth factor beta, TLR = Toll-like receptor, TNF-α = Tumor necrosis factor-alpha, TNFR = Tumor necrosis factor receptor, Rantes = Regulated upon Activation, Normal T Cell Expressed and Presumably Secreted, RBCs = Red blood cells).

**Figure 3 cancers-14-05656-f003:**
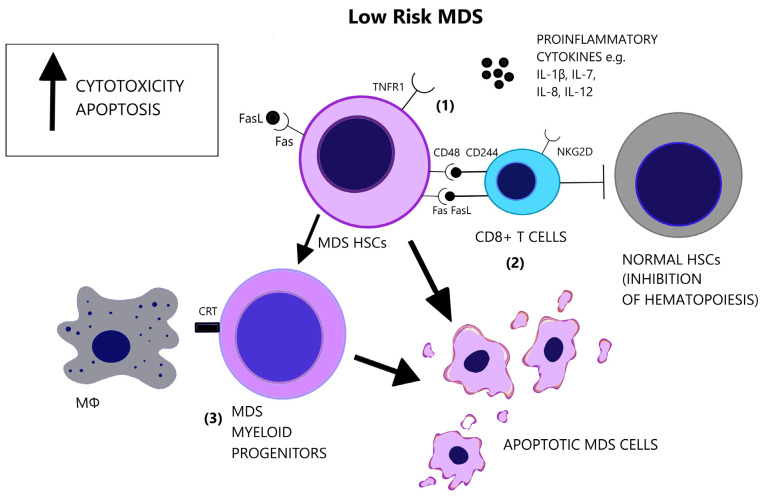
Low-risk MDS is characterized by a pro-apoptotic and cytotoxic microenvironment. (**1**) Apart from the elevated number of pro-inflammatory cytokines in the BM, there is increased responsiveness of the MDS cells that express the pro-apoptotic receptors, Fas and TNFR1. (**2**) CD8+ T-cells target MDS cells and exhibit augmented cytotoxicity, partly due to the expression of NKG2D and CD244 receptors. However, CD8+ T-cells suppress also normal hematopoietic cells and inhibit hematopoiesis. (**3**) MDS myeloid progenitor cells, derived from MDS HSCs, express the prophagocytic molecule CRT, possibly representing the main factor of cellular loss in low-risk MDS. (CRT = Calretinin, Fas = Fas receptor, FasL = Fas ligand, HSCs = Hematopoietic stem cells, IL-1β = Interleukin-1 beta, IL-7 = Interleukin-7, IL-8 = Interleukin-8, IL-12 = Interleukin-12, MΦ = Macrophage, NK2GD = Natural killer group 2D, TNFR1 = Tumor necrosis factor receptor 1).

**Figure 4 cancers-14-05656-f004:**
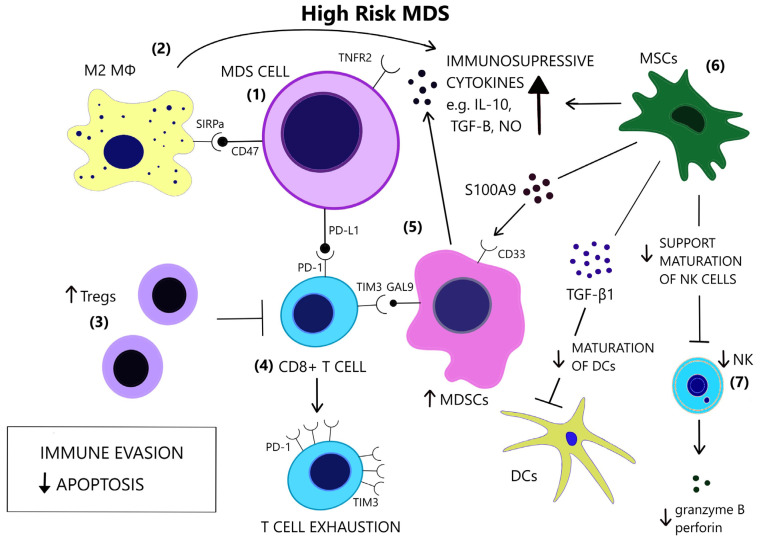
In contrast to low-risk MDS, immune evasion and decreased apoptosis enable clonal expansion in the BM of high-risk MDS. (**1**) In high risk MDS cells, there is a preferential expression of the anti-apoptotic TNFR2 and the anti-phagocytic CD47. (**2**) Macrophages acquire a M2 phenotype and produce IL-10. (**3**) Tregs expand and contribute to immune suppression. (**4**) CD8+ T-cell exhaustion is mediated by the expression of immune checkpoint molecules PD-1, PD-L1 and TIM3. (**5**) TIM3-ligand, Gal9, is expressed on MDSCs, which are induced upon binding of alarmin S100A9 on their CD33 receptor. MDSCs are increased and provide a large amount of immunosuppressive cytokines in the BM of high risk MDS. (**6**) MSCs are implicated in the anti-inflammatory microenvironment as they are a source of S100A9 and inhibitory molecules, such as TGF-β1, which suppresses DC maturation. (**7**) NK cells receive inadequate support from MSCs and exhibit impaired maturation, numerical and functional deficiency. (GAL9 = Galectin9, DCs = Dendritic cells, IL-10 = Interleukin-10, M2 MΦ = M2 macrophage, MDSCs = Myeloid-derived suppressor cells, MSCs = Mesenchymal stromal cells, NK = Natural killer, NO = Nitric oxide, PD-1 = Programmed cell death protein 1, PD-L1 = Programmed death-ligand 1, S100A9 = S100 calcium-binding protein A9, SIRPa = Signal regulatory protein alpha, TGF-β = Transforming growth factor-beta, TIM3 = T-cell immunoglobulin and mucin-domain containing 3, TNFR2 = Tumor necrosis factor receptor 2, Tregs = T regulatory cells).

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
