# Peer review of "Bone Marrow Immune Microenvironment in Myelodysplastic Syndromes"

_cancers, 2022, doi:10.3390/cancers14225656_

Round 1
Reviewer 1 Report
This is a very nice, well written, comprehensive review on various aspects of immune dysregulation in the BM microenvironment in MDS. It covers a big body of the literature on both innate and adaptive immune responses throughout all the spectrum, from normal hematopoiesis, aging, CHIP, to low and high risk MDS, and their role in disease initiation, bone marrow failure, immune evasion and progression. A few points :
1. Although the title of the section "5q- syndrome as a paradigm of immune dysregulation in MDS initiation" is nice and accurate, I feel it has been used in other reviews, too. I would recommend it should be changed, especially the word "paradigm".
2. A molecule with a significant role in MDS that has been left out of this review is HMGB1. HMGB1 has been repeatedly reported to be elevated in MDS and is now under clinical trial.
3. Another part that has not been covered at all is therapeutic approaches targeting dysregulated immune pathways in MDS- of course this itself could be the topic of a separate review, but I feel that either as an additional small paragraph/section, or throughout the text, where relevant, would elevate this manuscript. I don't consider this change necessary though.
Author Response
We sincerely thank the reviewer for the constructive comments, that help improve our manuscript. The following changes are made with respect to the reviewer's comments:
- In the title of paragraph 5.1 the word "paradigm" is substituted by "representative example".
- A paragraph (5.3) is added with a brief report on HMGB1, and its involvement in MDS.
- A brief paragraph (6) is inserted just before the conclusions, selectively refering to the therapeutic implications of targeting elements of the immune microenvironment in MDS.
Reviewer 2 Report
Review article is well-organized and written. The figures are extremely well done and complement the text nicely. Nice comprehensive review of the BM immune microenvironment.
Author Response
We thank the reviewer for the kind and generous comments.